# Highly Oxygenated Constituents from a Marine Alga-Derived Fungus *Aspergillus giganteus* NTU967

**DOI:** 10.3390/md18060303

**Published:** 2020-06-06

**Authors:** Jih-Jung Chen, Shih-Wei Wang, Yin-Ru Chiang, Ka-Lai Pang, Yueh-Hsiung Kuo, Tsai-Yen Shih, Tzong-Huei Lee

**Affiliations:** 1Faculty of Pharmacy, School of Pharmaceutical Sciences, National Yang-Ming University, Taipei 11221, Taiwan; chenjj@ym.edu.tw; 2Department of Medical Research, China Medical University Hospital, China Medical University, Taichung 404, Taiwan; 3Department of Medicine, Mackay Medical College, New Taipei City 25245, Taiwan; shihwei@mmc.edu.tw; 4Graduate Institute of Natural Products, College of Pharmacy, Kaohsiung Medical University, Kaohsiung 80708, Taiwan; 5Biodiversity Research Center, Academia Sinica, Taipei 11529, Taiwan; yinru915@gate.sinica.edu.tw; 6Institute of Marine Biology and Center of Excellence for the Oceans, National Taiwan Ocean University, Keelung 20224, Taiwan; klpang@ntou.edu.tw; 7Department of Chinese Pharmaceutical Sciences and Chinese Medicine Resources, China Medical University, Taichung 40447, Taiwan; kuoyh@mail.cmu.edu.tw; 8Department of Biotechnology, Asia University, Taichung 41354, Taiwan; 9Chinese Medical Research Center, China Medical University, Taichung 40447, Taiwan; 10Institute of Fisheries Science, National Taiwan University, Taipei 10617, Taiwan; r05b45006@ntu.edu.tw

**Keywords:** *Aspergillus giganteus*, Trichocomaceae, bioactive natural products, Polyketides, aspergilsmin

## Abstract

Agar-based disc diffusion antimicrobial assay has shown that the ethyl acetate extract of the fermented broth of *Aspergillus*
*giganteus* NTU967 isolated from *Ulva lactuca* exhibited significant antimicrobial activity in our preliminary screening of bioactive fungal strains. Therefore, column chromatography of the active principles from liquid- and solid–state fermented products of the fungal strain was carried out, and which had led to isolation of eleven compounds. Their structures were determined by spectral analysis to be seven new highly oxygenated polyketides, namely aspergilsmins A–G (**1**–**7**), along with previously reported patulin, deoxytryptoquivaline, tryptoquivaline and quinadoline B. Among these, aspergilsmin C (**3**) and patulin displayed promising anticancer activities against human hepatocellular carcinoma SK-Hep-1 cells and prostate cancer PC-3 cells with IC_50_ values between 2.7–7.3 μM. Furthermore, aspergilsmin C (**3**) and patulin exhibited significant anti-angiogenic functions by impeding cell growth and tube formation of human endothelial progenitor cells without any cytotoxicity.

## 1. Introduction

The so called marine-derived fungi have been isolated from a wide array of marine organisms such as mangroves, algae, sponges and corals, whose habitats distribute from deep sea to intertidal zone. Among these, algae-derived fungi have been reported to be the largest source of secondary metabolites with diversified bioactivities [1,2], and that can be exploited potentially as lead compounds for new drug development. It was reported that by employing one of the post-genomic strategies, one strain many compounds (OSMAC), on the cultivation of the fungal strains could enhance the quantity and diversity of fungal secondary metabolites [3,4]. The OSMAC approach usually involved the manipulation of culturing parameters, such as media formulation, temperature, agitation, luminosity, aeration, etc. [5,6,7]. In addition. easy to scale-up and quality control of the fungal metabolites would be another major advantage that made fungi to be one of the best options for natural product research and new lead discovery.

Taiwan is an island located at tropical and subtropical region with highly diversified marine algal species [8], indicating an abundant resource of the fungal endophytes. However, the chemical investigation on the local algae-derived fungal strains are still rare so far. Thus, an efficient agar-based disc diffusion assay was applied for the preliminary biological screening against *Escherichia coli*, *Staphylococcus aureus*, *Candida albicans* and *Cryptococcus neoformans* [9], the ethyl acetate extracts of fermented broths of *Aspergillus giganteus* NTU967 derived from the green alga *Ulva lactuca* were found to exhibit significant inhibition zone against *S. aureus* and *C. neoformans*. Therefore, chemical investigation on both the fermented products of *Aspergillus giganteus* NTU967 was performed, and this has resulted in the isolation and identification of seven previously unreported highly oxygenated polyketides **1**–**7** (Figure 1) together with four known compounds. This study describes the isolation and characterization of the new compounds together with their bioactivities (see Appendix A).

## 2. Results and Discussion

### 2.1. Isolation and Characterization of Secondary Metabolites

In this study, the green alga *Ulva lactuca*-derived fungal strain *Aspergillus giganteus* NTU967 was cultured in both solid- and liquid–state culturing conditions in order to enrich the diversity of the fungal secondary metabolites and eleven chemical entities including seven new compounds **1**–**7** and four previously reported compounds, patulin, deoxytryptoquivaline, tryptoquivaline and quinadoline B, were obtained from the fermented products. Of the known compounds isolated, patulin, a highly oxygenated C_7_ mycotoxin, with a hemiacetal functionality is apt to racemize naturally to form a pair of enantiomers and was first characterized in 1943 under the name of tercinin as a potential antimicrobial agent [10]. Recently, patulin in combination with oxaliplatin were found to exhibit synergism against human colorectal cancer [11]. Deoxytryptoquivaline, tryptoquivaline and quinadoline—three quinazolone-containing alkaloids—were identified by comparison of spectroscopic data with literatures [12,13]. In addition to be isolated from *Aspergillus* spp., a series of tryptoquivaline analogs have ever been obtained from a marine sea fan-derived fungus *Neosartorya siamensis* [14].

Compound **1**, obtained as colorless oil, was determined to have a molecular formula of C_9_H_12_O_5_, as evidenced by its ^13^C NMR spectrum (Table 1) and a pseudomolecular ion [M + Na]^+^ at *m*/*z* 223.0574 (calcd. 223.0582 for C_9_H_12_O_5_Na) in the positive mode of HRESIMS analysis. The IR absorptions at 1745 and 1672 revealed the presence of an ester carbonyl and a conjugated ketone functionality, respectively. The ^1^H NMR (CD_3_OD, 400 MHz) of compound **1** revealed two oxygen-bearing three-proton signals at δ_H_ 3.35 (H_3_-8) and 3.71 (H_3_-9), one methylene signal at δ_H_ 2.61 (d, *J* = 6.2 Hz, H_2_-2), one oxygen-bearing methylene signal at δ_H_ 4.55 (H_2_-7), one oxygen-bearing methine signal at δ_H_ 4.55 (H-3) and an olefinic methine signal at δ_H_ 7.66 (s, H-6) (Table 2). The ^13^C NMR (CD_3_OD, 125 MHz) coupled with phase-sensitive HSQC spectrum of compound **1** showed nine signals including two methoxyl carbons at δ_C_ 57.7 (C-8) and 52.7 (C-9), one methylene carbon at δ_C_ 36.7 (C-2), one oxymethylene carbon at δ_C_ 75.9 (C-7), one oxygenated methine carbon at δ_C_ 69.7 (C-3), one oxygenated olefinic carbon at δ_C_ 166.1 (C-6), one nonprotonated olefinic carbon at δ_C_ 116.2 (C-5), one ester carbonyl carbon at δ_C_ 172.9 (C-1) and one conjugated ketone carbon at δ_C_ 191.8 (C-4) (Table 1). On account of the molecular formula C_9_H_12_O_5_, the double bond equivalent (DBE) of compound **1** was four including a double bond and two carbonyl groups. Thus, there should be one additional ring in compound **1**. Further two dimensional NMR analysis including one cross-peak of δ_H_ 2.61 (H_2_-2)/δ_H_ 4.55 (H-3) in the COSY spectrum of compound **1** in combination with key correlations of δ_H_ 2.61 (H_2_-2)/δ_C_ 69.7 (C-3) and 191.8 (C-4); δ_H_ 4.55 (H-3)/δ_C_ 172.9 (C-1) and 191.8 (C-4); δ_H_ 7.66 (H-6)/δ_C_ 69.7 (C-3), 191.8 (C-4), 116.2 (C-5) and 75.9 (C-7); δ_H_ 4.55 (H_2_-7)/δ_C_ 191.8 (C-4); δ_H_ 3.35 (H_3_-8)/δ_C_ 75.9 (C-7); δ_H_ 3.71 (H_3_-9)/δ_C_ 172.9 (C-1) in the HMBC spectrum of compound **1** (Figure 2), corroborated the gross structure of **1**. The optical rotation value of compound **1** ([α] ^27^_D_ −0.36) was close to zero, revealing that compound **1** could be a racemate.

The molecular formula of **2**, C_9_H_12_O_5_, was deduced through analysis of its ^13^C NMR and HRESIMS data. Its IR spectrum revealed the presence of an ester carbonyl (1737 cm^−1^) and a conjugated ketone group (1673 cm^−1^). The ^1^H NMR spectrum of compound **2** coupled with phase-sensitive HSQC spectrum showed two methoxyl signals at δ_H_ 3.45 (s, H_3_-8) and 3.67 (s, H_3_-9), one methylene signal at δ_H_ 3.14 (s, H_2_-2), one oxygenated methylene signal at δ_H_ 4.49 (d, *J* = 4.7 Hz, H_2_-6), one carbinoyl methine signal at δ_H_ 3.70 (t, *J* = 4.7 Hz, H-5), and one olefinic methine signal at δ_H_ 7.50 (s, H-7) (Table 2). Nine carbon resonances including two methoxyl at δ_C_ 58.7 (C-8) and 52.5 (C-9), one oxygenated methylene at δ_C_ 72.3 (C-6), one methylene at δ_C_ 31.1 (C-2), one olefinic methine at δ_C_ 164.0 (C-7), one carbinoyl methine at δ_C_ 77.3 (C-5), one nonprotonated olefinic carbon at δ_C_ 111.7 (C-3), one ester carbonyl carbon at δ_C_ 173.6 (C-1), and one ketone carbon at δ_C_ 191.3 were observed in the ^13^C NMR of compound **2** (Table 1), which were supported by phase-sensitive HSQC spectrum. On account of the molecular formula of **2**, C_9_H_12_O_5_, compound **2** would contain a ring in addition to a double bond, a ketone and an ester carbonyl to fit the unsaturation number. Further comprehensive analysis of two dimensional NMR spectra of compound **2** (Figure 2), one cross-peak of δ_H_ 3.70 (H-5)/δ_H_ 4.49 (H_2_-6) in the COSY spectrum together with key cross-peaks of δ_H_ 3.14 (H_2_-2)/δ_C_ 111.7 (C-3); δ_H_ 4.49 (H_2_-6)/δ_C_ 191.3 (C-4); δ_H_ 7.50 (H-7)/δ_C_ 31.3 (C-2), 111.7 (C-3), 191.3 (C-4) and 72.3 (C-6); δ_H_ 3.45 (H_3_-8)/δ_C_ 77.3 (C-5); δ_H_ 3.67 (H_3_-9)/δ_C_ 173.6 (C-1) in the HMBC spectrum established the structure of **2**.

The physical and NMR data of compound **3** and compound **4** were almost compatible with those of patulin except that an additional methyl and an additional ethyl were observed in the ^1^H and ^13^C NMR spectra of compound **3** and compound **4** (Table 1 and Table 2), respectively. The pseudomolecular ion [M + H]^+^ at *m*/*z* 169.0493 and 183.0655 in the HRESIMS of compounds **3** and **4**, 14 and 28 Da more than that of patulin, confirmed that compound **3** and compound **4** were the methyl and ethyl derivatives of patulin, respectively. In the HMBC spectra of compounds **3** and **4**, distinctive cross-peaks of δ_H_ 3.54 (H_3_-8)/δ_C_ 95.9 (C-7) and δ_H_ 3.71 and 3.91 (H_2_-8)/δ_C_ 94.8 (C-7) indicated the methyl and ethyl groups were attached to C-7 of compounds **3** and **4**, respectively. The structures of compound **3** and compound **4** were thus elucidated to be as shown.

Compound **5**, obtained as colorless oil, was determined to have a molecular formula of C_10_H_14_O_5_, as evidenced by its HRESIMS analysis and ^13^C NMR spectrum (Table 1). It contained a hydroxy, a γ-lactone carbonyl and a double bond due to the IR absorption bands at 3435, 1768 and 1643 cm^−1^, respectively. Interpretations of the ^1^H NMR data along with the HSQC spectrum of compound **5** showed two methyl signals at δ_H_ 3.36 (s, H_3_-10) and 1.24 (t, *J* = 7.1 Hz, H_3_-10), two oxygenated methylene signals at δ_H_ 4.40 (d, *J* = 6.9 Hz, H_2_-6) and 3.63 (m, H_2_-8), two olefinic methine signals at δ_H_ 6.31 (s, H-2) and 5.85 (t, *J* = 6.9 Hz, H-5) and one dioxygenated methine signal at δ_H_ 5.52 (s, H-7) (Table 2). The ^13^C NMR data of compound **5** coupled with its phase-sensitive HSQC assignments showed one methoxyl carbon at δ_C_ 53.2 (C-10), one methyl carbon at δ_C_ 15.4 (C-9), two oxygenated methylene carbons at δ_C_ 62.8 (C-8) and 57.3 (C-6), two olefinic methine carbons at δ_C_ 119.7 (C-2) and 115.3 (C-5), one dioxygenated methine carbon at δ_C_ 97.6 (C-7) and three nonprotonated carbons at δ_C_ 169.6 (C-1), 156.5 (C-3) and 148.2 (C-4) (Table 1). Correlations of δ_H_ 5.85 (H-5)/δ_H_ 4.40 (H_2_-6) and δ_H_ 3.63 (H_2_-8)/δ_H_ 1.24 (H_3_-9) in the COSY spectrum of compound **5** accompanied with key correlations including δ_H_ 6.31 (H-2)/δ_C_ 169.6 (C-1), 156.5 (C-3), 148.2 (C-4) and 97.6 (C-7); δ_H_ 5.58 (H-5)/δ_C_ 156.5 (C-3) and 148.2 (C-4); δ_H_ 3.63 (H_2_-8)/δ_C_ 97.6 (C-7); δ_H_ 3.36 (H-10)/δ_C_ 97.6 (C-7) in the HMBC spectrum of compound **5** (Figure 2), established the gross structure of **5**. The configuration of Δ^4^ in compound **5** was determined to be *E* form based on a key correlation of δ_H_ 4.40 (H_2_-6)/δ_H_ 5.52 (H-7) in the ROESY of compound **5** (Figure 2). Since compound **5** had an acetal carbon at its C-7 and an optical rotation value close to zero ([α] ^27^_D_ +0.02), compound **5** was deduced to be an acetal racemate.

The ^1^H NMR data of compounds **6** and **7** were almost identical with that of compound **3** except that an olefinic proton at δ_H_ 6.02 (H-5) in compound **3** was substituted by a methylene group at δ_H_ 1.91 and 2.38 (H_2_-5) in compound **6** and δ_H_ 1.99 and 2.34 (H_2_-5) in compound **7** and an additional methoxyl functionality at δ_H_ 3.23 (H_3_-9) and δ_H_ 3.19 (H_3_-9) was observed in compound **6** and compound **7** (Table 2), respectively. These changes also reflected in their ^13^C NMR data (Table 1), in which two olefinic signals at δ_C_ 150.9 (C-4) and 109.3 (C-5) in compound **3** were replaced by a ketal carbon signal (δ_C-4_ 107.6 in **6**; δ_C-4_ 107.2 in **7**) along with a methylene carbon resonance (δ_C-5_ 40.3 in **6**; δ_C-5_ 41.9 in **7**). The additional methoxyl groups in compounds **6** and **7** were deduced to be attached at C-4 due to key cross-peaks of H_3_-9/C-4 in the HMBC spectra of both compound **6** and compound **7** (Figure 2). In the ROESY spectra of compounds **6** and **7**, a key correlation of δ_H_ 3.23 (H_3_-9)/δ_H_ 5.11 (H-7) confirmed that two methoxyls in compound **6** were oriented in different phases, while a key cross-peak of δ_H_ 3.46 (H_3_-8)/δ_H_ 3.19 (H_3_-9) indicated that two methoxyls in compound **7** were located on the same side (Figure 2). The structures of compounds **6** and **7** were thus established as shown in Figure 1.

### 2.2. Anticancer and Anti-Angiogenic Assays of Secondary Metabolites

All eleven pure isolates were subjected to biological assays. Among these, compound **3** exerted promising anticancer activities against human hepatocellular carcinoma SK-Hep-1 cells and prostate cancer PC-3 cells with IC_50_ values of 2.7 ± 0.2 and 7.3 ± 0.3 μM (Table 3). Paclitaxel, a well-known anticancer agent, was used as the positive control. Additionally, we evaluated the anti-angiogenic activities of all the pure isolates against human endothelial progenitor cells (EPCs). As shown in Table 3, compound **3** and patulin exhibited most potent anti-angiogenic activities by suppressing EPCs growth with IC_50_ values of 4.6 ± 0.3 and 4.7 ± 0.2 μM, respectively. Since capillary-like tubules are the essential characteristic of angiogenesis, we next performed tube formation assay to validate the anti-angiogenic effects of compound **3** and patulin in EPCs with sorafenib as a positive control. The results showed that compound **3** and patulin concentration-dependently inhibited capillary tube formation of EPCs (Figure 3A,B). Furthermore, it was found that compound **3** and patulin did not induce the release of lactate dehydrogenase (LDH) in EPCs (Figure 3C), suggesting that these two compounds display anti-angiogenesis property without the cytotoxic fashion. Compound **4** bearing an ethoxyl group at its C-7 instead of a hydroxy and a methoxyl at C-7 of patulin and **3**, respectively, reduced its bioactivity significantly. It was thus speculated that the size of the functional group attached at C-7 and an olefinic functionality at C-4 of patulin could play crucial roles in the anticancer and anti-angiogenic activities. These findings provide evidences that both compound **3** and patulin may serve as the potential natural products to block tumor angiogenesis for cancer treatment.

## 3. Materials and Methods

### 3.1. General Experimental Procedures

Optical rotations and UV were measured on a JASCO *P-*2000 polarimeter (Tokyo, Japan) and Thermo UV-Visible Heλios α Spectrophotometer (Bellefonte, CA, USA), respectively. ^1^H and ^13^C NMR were acquired on Bruker AVIII HD 400 and Bruker AVIII-500 spectrometer (Ettlingen, Germany). Low and high resolution mass spectra were obtained using an API4000 triple quadrupole mass spectrometer (Applied Biosystems, Foster City, CA, USA) and Q Exactive Plus Hybrid Quadrupole-Orbitrap Mass Spectrometer (Thermo Fisher Scientific, Bremen, Germany), respectively. IR spectra were recorded on a JASCO FT/IR 4100 spectrometer (Tokyo, Japan). Sephadex LH-20 (GE Healthcare, Uppsala, Sweden) and Diaion HP-20 (Mitsubishi Chemical, Tokyo, Japan) was used for open column chromatography. An HPLC pump L-7100 (Hitachi, Japan) equipped with a refractive index detector (Bischoff, Leonberg, Germany) was used for compound purification. All the organic solvents were purchased from Merck (Darmstadt, Germany).

### 3.2. Fungal Strain and Culture

*Aspergillus giganteus* NTU967 was isolated from the marine green alga *Ulva lactuca* collected from the northeast coast of Taiwan and was identified by sequencing of the internal transcribed spacer regions of the rDNA (ITS) and β-tubulin gene. A BLAST search of the ITS sequence (GenBank accession no. MH250052) was not conclusive and led to the best matches as *Aspergillus clavatoanicus*, *A. clavatus*, *A. giganteus* and *A. longivesica* (query coverage 94–100%, identity 98–99%) while a BLAST search of the β-tubulin gene resulted the best matches as *A. giganteus* (query coverage 97–100%, identity 98.48–99.63%). For liquid culture, the mycelium of *Aspergillus giganteus* NTU967 was inoculated into 5 L serum bottles, each containing 2 g Peptone (Becton, Dickinson and Company, Sparks, MD, USA), 1 g yeast extract (Becton, Dickinson and Company, Sparks, MD, USA), 10 g Dextrose (Becton, Dickinson and Company, Sparks, MD, USA) and 2.5 L deionized water. The fermentation was conducted with aeration at 25–30 °C for 16 days. For solid culture, the mycelium of *Aspergillus giganteus* NTU967 was inoculated into 500 mL flasks, each containing 50 g brown rice (Santacruz, Taiwan), 2% yeast extract (Becton, Dickinson and Company, Sparks, MD, USA), 1% sodium tartrate and 1% KH_2_HPO_4_ in 20 mL deionized water. The solid culture was conducted at 25–30 °C for 30 days.

### 3.3. Extraction and Isolation of Secondary Metabolites

For liquid culture, the filtered fermented broth (15.0 L) of *Aspergillus giganteus* NTU967 was partitioned three times with 30 L EtOAc, then concentrated in vacuum to dryness (8.0 g). Subsequently, the crude extract was redissolved in 20 mL MeOH, then applied onto a Sephadex LH-20 column (2.5 cm i.d. × 68 cm) eluted with MeOH at a flow rate of 2.5 mL/min. Each fraction (20 mL) collected was checked for its compositions by TLC using CH_2_Cl_2_-MeOH (10:1, *v*/*v*) for development, and dipping in vanillin–H_2_SO_4_ was used in the detection of compounds with similar skeletons. All the fractions were combined into four samples I–VI. Subsequently, sample III with antimicrobial activity was precoated with 15.0 g Diaion HP-20 gel, then applied onto a Diaion HP-20 column (4.5 cm i.d. × 30 cm) eluted with mixtures of H_2_O/MeOH in a stepwise gradient mode with a flow rate of 2.0 mL/min to obtain four subsamples I–IV. Subsample III eluted by 75% MeOH was rechromatographed on a semipreparative reversed-phase column (Phenomenex Luna 5 μ PFP, 10 × 250 mm) with 35% MeOH_aq_ as eluent, 2 mL/min, to afford **1** (30.4 mg, *t_R_* = 13.5 min), **2** (15.2 mg, *t_R_* = 16.9 min) and **3** (27.4 mg, *t_R_* = 27.5 min). Subsample III was further purified on a semipreparative reversed-phase column (Thermo Hypersil 5 μ C_18_, 10 × 250 mm) with 25% MeCN_aq_ as eluent, 2 mL/min, to give **4** (22.8 mg, *t_R_* = 35.1 min), **5** (18.4 mg, *t_R_* = 23.4 min), **6** (16.6 mg, *t_R_* = 26.1 min) and **7** (35.6 mg, *t_R_* = 27.6 min).

For solid–state culture, the fermented products were lyophilized, ground into powder (750 g) and extracted three times with equal volumes of methanol. Extracts were first partitioned with *n*-hexane and the methanol layers suspended in deionized H_2_O, then partitioned with ethyl acetate and concentrated to obtain dried ethyl acetate extract (7.0 g). For compound separation, the ethyl acetate extract was subjected to Sephadex LH-20 column chromatography (2.5 i.d. × 68.0 cm), using methanol as the eluent at a flow rate of 2.5 mL/min to give 30 fractions (20.0 mL/fr.). All the fractions were combined into 6 samples as I–VI based on the results of TLC analysis and antimicrobial assay. Sample III with antimicrobial activity was precoated with 20.0 g Diaion HP-20 gel, then applied onto a Diaion HP-20 column (4.5 cm i.d. × 30 cm) eluted with mixtures of H_2_O/MeOH in a stepwise gradient mode with a flow rate of 2.0 mL/min to get four subsamples I–IV. Subsample II eluted by 50% MeOH was rechromatographed on a semipreparative reversed-phase column (BIOSIL Pro-ODS-U 5 μ, 10 × 250 mm) with 15% MeOH_aq_ as eluent, 2 mL/min, to obtain patulin (16.7 mg, *t_R_* = 20.1 min). Subsample IV eluted by 100% MeOH was further purified on a semipreparative reversed-phase column (Phenomenex Luna 5 µ PFP, 10 × 250 mm) with 75% MeOH_aq_ as eluent, 2 mL/min, to afford quinadoline B (27.7 mg, *t_R_* = 11.8 min), deoxytryptoquivaline (21.0 mg, *t_R_* = 23.0 min) and tryptoquivaline (16.4 mg, *t_R_* = 36.2 min).

*Aspergilsmin A* (**1**): Colorless oil; [α] ^27^_D_ −0.36 (*c* = 0.05, MeOH); IR (ZnSe) ν_max_: 2951, 1745, 1672, 1611, 1456, 1438, 1399, 1333, 1308, 1283, 1254, 1196, 1171, 1105, 1055, 1033 and 1009 cm^−1^; UV λ_max_ (MeOH) (log ε) 261 (3.9) nm; ^1^H and ^13^C NMR spectroscopic data: see Table 1 and Table 2; HRESIMS [M + Na]^+^ at *m*/*z* 223.0574 (calcd. 223.0582 for C_9_H_12_O_5_Na).

*Aspergilsmin B* (**2**): Colorless oil; [α] ^27^_D_ +1.22 (*c* = 0.05, MeOH); IR (ZnSe) ν_max_: 2947, 1737, 1673, 1619, 1443, 1406, 1344, 1291, 1257, 1157, 1097, 1056, 1043, 1024, 1011 and 854 cm^−1^; UV λ_max_ (MeOH) (log ε) 268 (3.9) nm; ^1^H and ^13^C NMR spectroscopic data: see Table 1 and Table 2; HRESIMS [M + Na]^+^ at *m*/*z* 223.0573 (calcd. 223.0582 for C_9_H_12_O_5_Na).

*Aspergilsmin C* (**3**): Colorless oil; [α] ^27^_D_ −3.52 (*c* = 0.05, MeOH); IR (ZnSe) ν_max_: 2955, 1780, 1536, 1443, 1406, 1344, 1210, 1065 and 868 cm^−1^; UV λ_max_ (MeOH) (log ε) 274 (4.0) nm; ^1^H and ^13^C NMR spectroscopic data: see Table 1 and Table 2; HRESIMS [M + H]^+^ at *m*/*z* 169.0493 (calcd. 169.0501 for C_8_H_9_O_4_).

*Aspergilsmin D* (**4**): Colorless oil; [α] ^27^_D_ −2.14 (*c* = 0.05, MeOH); IR (ZnSe) ν_max_: 2945, 1780, 1635, 1404, 1092 and 1019 cm^−1^; UV λ_max_ (MeOH) (log ε) 275 (4.0) nm; ^1^H and ^13^C NMR spectroscopic data: see Table 1 and Table 2; HRESIMS [M + H]^+^ at *m*/*z* 183.0655 (calcd. 183.0657 for C_9_H_11_O_4_).

*Aspergilsmin E* (**5**): Colorless oil; [α] ^27^_D_ +0.02 (*c* = 0.05, MeOH); IR (ZnSe) ν_max_: 3435, 1768, 1643, 1053 and 1008 cm^−1^; UV λ_max_ (MeOH) (log ε) 223 (3.7) and 270 (4.0) nm; ^1^H and ^13^C NMR spectroscopic data: see Table 1 and Table 2; HRESIMS [M + H]^+^ at *m*/*z* 215.0915 (calcd. 215.0947 for C_10_H_15_O_5_).

*Aspergilsmin F* (**6**): Colorless oil; [α] ^27^_D_ −0.53 (*c* = 0.05, MeOH); IR (ZnSe) ν_max_: 2951, 1768, 1643, 1456, 1396, 1207, 1163, 1082, 1022 and 972 cm^−1^; UV λ_max_ (MeOH) (log ε) 257 (4.1) nm; ^1^H and ^13^C NMR spectroscopic data: see Table 1 and Table 2; HRESIMS [M + Na]^+^ at *m*/*z* 223.0579 (calcd. 223.0582 for C_9_H_12_O_5_Na).

*Aspergilsmin G* (**7**): Colorless oil; [α] ^27^_D_ −1.86 (*c* = 0.05, MeOH); IR (ZnSe) ν_max_: 2971, 2842, 1766, 1456, 1435, 1358, 1329, 1266, 1205, 1196, 1171, 1129, 1041, 1034 and 915 cm^−1^; UV λ_max_ (MeOH) (log ε) 271 (4.0) nm; ^1^H and ^13^C NMR spectroscopic data: see Table 1 and Table 2; HRESIMS [M + Na]^+^ at *m*/*z* 223.0574 (calcd. 223.0582 for C_9_H_12_O_5_Na).

### 3.4. Cell Culture

The human hepatocellular carcinoma cell line SK-Hep-1 and hormone refractory prostate cancer cell line PC-3 were purchased from the American Type Cell Culture Collection (Manassas, VA, USA) and maintained in DMEM medium containing 10% fetal bovine serum (FBS), penicillin (100 units/mL) and streptomycin (100 μg/mL). Cells were maintained in humidified air containing 5% CO_2_ at 37 °C. All cell culture reagents were purchased from Gibco-BRL life technologies (Grand Island, NY, USA). The isolation and maintenance of human CD34-positive endothelial progenitor cells (EPCs) were conducted using the standard method as previously described [15].

### 3.5. Biologic Assay for Anticancer Activity

SK-Hep-1 and PC-3 cancer cells were seeded onto 96-well plates in a density of 5 × 10^3^ cells per well. Overnight, cells were treated with the tested compounds for 48 h. Then, anticancer activity was determined by the SRB assay according to previously described procedures [16].

### 3.6. Biologic Assay for Anti-Angiogenic Activity

For EPCs’ cell growth assay, EPCs were cultured in 96-well plates at a density of 5 × 10^3^ cells in each well. Overnight, the culture medium was replaced with MV2 complete medium containing 2% FBS in the presence of the tested compounds for 48 h. The reaction was terminated after 48 h of incubation with 50% TCA. After the TCA fixation, every well was incubated for 15 min incubation with 0.4% sulforhodamine B in 1% acetic acid. The plates were then washed before the dye was dissolved by 10-mM Tris buffer. Absorbance density values were read by an enzyme-linked immunosorbent assay (ELISA) reader (515 nm).

For EPCs’ tube formation assay, EPCs were seeded with the density of 1.25 × 10^4^ cells per well in Matrigel-coated 96-well plates and incubated in an MV2 complete medium containing 2% FBS and the tested compounds for 24 h. EPCs differentiation and capillary-like tube formation was taken with the inverted phase contrast microscope. The long axis of each tube was measured with MacBiophotonics Image J software in 3 randomly chosen fields per well.

For EPCs’ cytotoxicity assay, EPCs (5 × 10^3^ cells/well) were seeded onto 96-well plates and incubated with MV2 complete medium containing 2% FBS and the tested compounds for 24 h. Then, the quantification of LDH release in the medium was done with a cytotoxicity assay kit.

## 4. Conclusions

In this report, we have identified seven new polyketides **1**–**7** along with four known compounds from a marine algicolous fungal strain *Aspergillus giganteus* NTU967. Of the compounds identified, compound **3** and its known analog patulin exhibited promising anticancer as well as significant anti-angiogenic activities when compared with the clinically used drugs.

## Figures and Tables

**Figure 1 marinedrugs-18-00303-f001:**
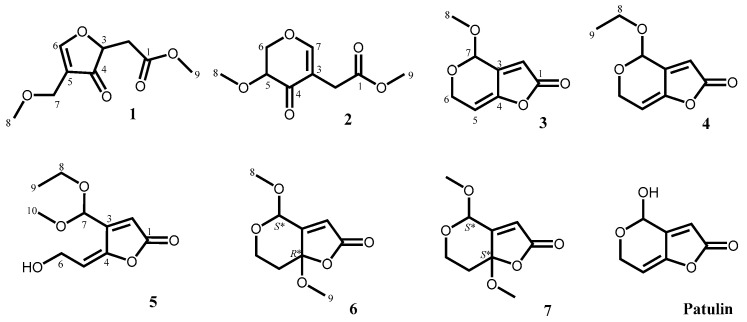
Chemical structures of compounds **1**–**7** and patulin.

**Figure 2 marinedrugs-18-00303-f002:**
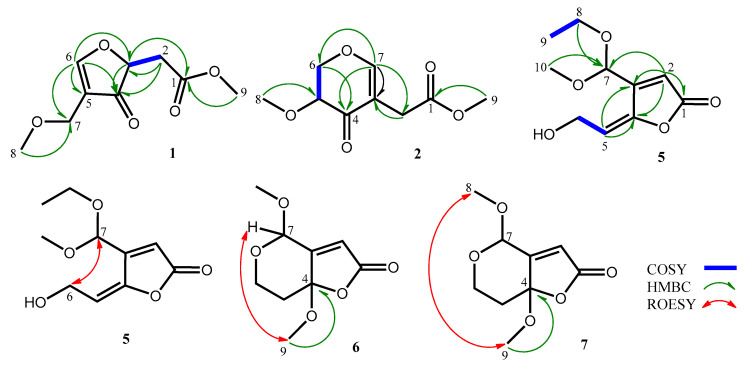
COSY and key HMBC and ROESY correlations of compounds **1**, **2**, **5**, **6** and **7**.

**Figure 3 marinedrugs-18-00303-f003:**
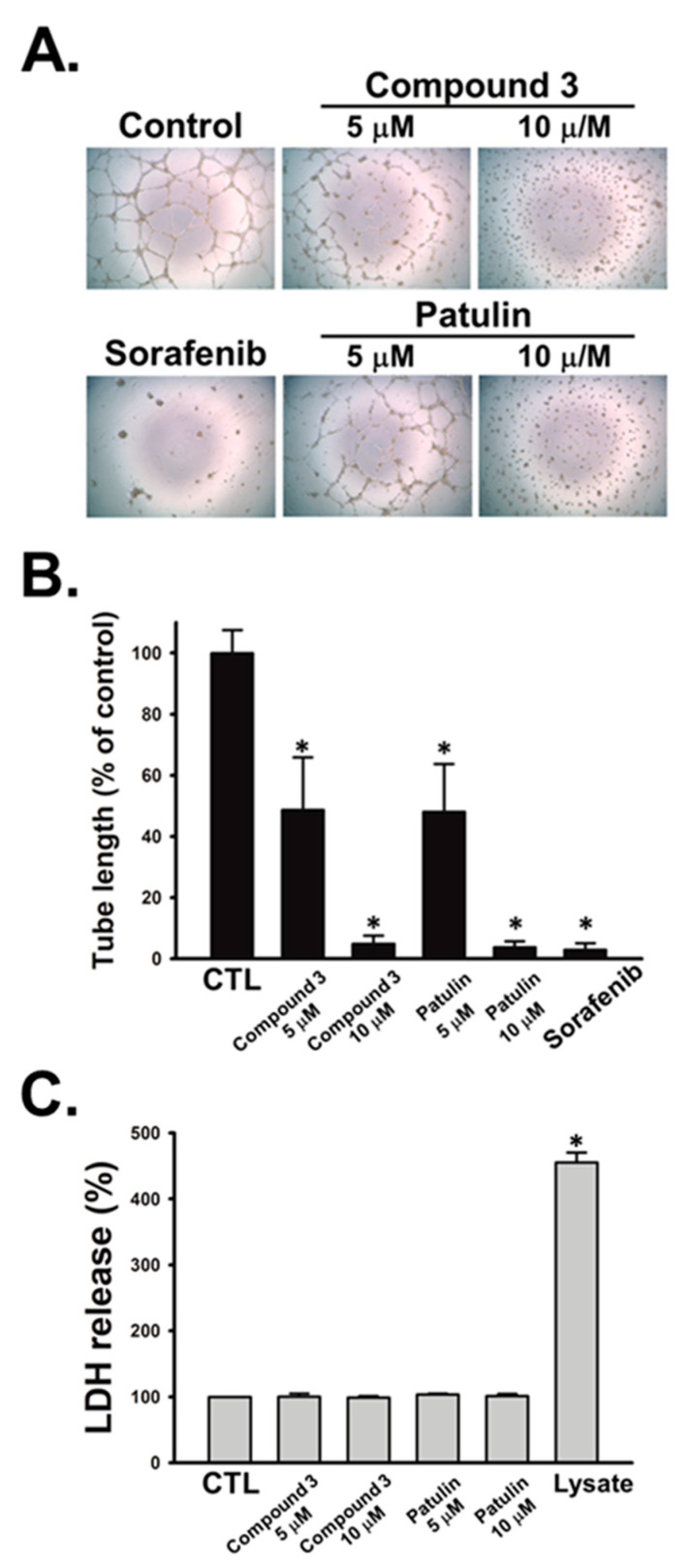
Effects of compound **3** and patulin on tube formation and cytotoxicity of human EPCs. (**A**) EPCs were treated with compound **3**, patulin and sorafenib (10 μM) for 24 h. Tubular morphogenesis was recorded by the inverted phase contrast microscope. Representative images of EPCs’ tube formation were shown; (**B**) tube formation was quantified by measuring the length of tubes using ImageJ software; (**C**) Cells were treated with the indicated compounds for 24 h, then the cytotoxicity was determined using LDH assay Data represent the mean ± S.E.M. of 4 independent experiments. * *p* < 0.05 compared with the control group.

**Table 1 marinedrugs-18-00303-t001:** ^13^C NMR spectroscopic data for compounds **1**–**7** (δ in ppm, mult.).

No.	1 *^a,b^*	2 *^a,b^*	3 *^a,b^*	4 *^a,b^*	5 *^a,b^*	6 *^a,b^*	7 *^a,b^*
1	172.9 s	173.6 s	170.6 s	170.6 s	169.6 s	170.6 s	170.6 s
2	36.7 t	31.1 t	111.9 d	111.1 d	119.7 d	117.1 d	119.9 d
3	69.7 d	111.7 s	147.5 s	147.6 s	156.5 s	163.9 s	160.9 s
4	191.8 s	191.3 s	150.9 s	151.3 s	148.2 s	107.6 s	107.2 s
5	116.2 s	77.3 d	109.3 d	109.4 d	115.3 d	40.3 t	41.9 t
6	166.1 d	72.3 t	60.0 t	60.0 t	57.3 t	62.3 t	58.3 t
7	75.9 t	164.0 d	95.9 d	94.8 d	97.6 d	98.8 d	97.4 d
8	57.7 q	58.7 q	56.5 q	65.6 t	62.8 t	57.3 q	55.5 q
9	52.7 q	52.5 q		15.4 q	15.4 q	51.2 q	52.2 q
10					53.2 q		

***^a^*** Measured in CD_3_OD (125 MHz); ***^b^*** Multiplicties were obtained from phase-sensitive HSQC experiments.

**Table 2 marinedrugs-18-00303-t002:** ^1^H NMR spectroscopic data for compounds **1**–**7** (δ in ppm, mult., *J* in Hz).

No.	1 *^a^*	2 *^b^*	3 *^b^*	4 *^b^*	5 *^b^*	6 *^b^*	7 *^b^*
1							
2	2.61 d (6.2)	3.14 s	6.60 s	6.05 s	6.31 s	6.12 s	6.23 s
3	4.55 *^c^*						
4							
5		3.70 t (4.7)	6.02 m	6.02 m	5.85 t (6.9)	1.91 m	1.99 m
						2.38 dt (13.5, 2.1)	2.34 d (13.2)
6	7.66 s	4.49 d (4.7)	4.37 dd (17.5, 4.4)	4.36 dd (17.3, 4.4)	4.40 d (6.9)	3.76 td (12.2, 2.1)	3.71 m
			4.56 dd (17.5, 2.5)	4.56 dd (17.3, 2.6)		4.01 m	4.10 td (12.0, 2.0)
7	4.55 *^c^*	7.50 s	5.68 s	5.79 s	5.52 s	5.11 s	5.52 s
8	3.35 s	3.45 s	3.54 s	3.71 m	3.63 m	3.60 s	3.46 s
				3.91 m			
9	3.71 s	3.67 s		1.26 t (7.1)	1.24 t (7.1)	3.23 s	3.19 s
10					3.36 s		

*^a^* Measured in CD_3_OD (400 MHz); *^b^* Measured in CD_3_OD (500 MHz); ^c^ Signals were overlapped and were picked up from HSQC spectrum.

**Table 3 marinedrugs-18-00303-t003:** Anticancer and anti-angiogenic activities of compound **3** and patulin.

Compounds	Anticancer (IC_50_, μM)		Anti-Angiogenesis (IC_50_, μM)
	SK-Hep-1 *^a^*	PC-3 *^b^*	EPC *^c^*
Compound **3**	2.7 ± 0.2	7.3 ± 0.3	4.6 ± 0.3
Patulin	2.9 ± 0.1	2.7 ± 0.1	4.7 ± 0.2
Paclitaxel ***^d^***	0.011 ± 0.002	0.013 ± 0.002	–
Sorafenib ***^d^***	–	–	4.8 ± 0.3

***^a^*** Hepatocellular carcinoma cells; ***^b^*** Prostate cancer cells; ***^c^*** Endothelial progenitor cells; ***^d^*** Positive control.

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
