# Peer review of "Highly Oxygenated Constituents from a Marine Alga-Derived Fungus Aspergillus giganteus NTU967"

_marinedrugs, 2020, doi:10.3390/md18060303_

Round 1
Reviewer 1 Report
The manuscript entitled “Highly Oxygenated Constituents from A Marine Alga-derived Fungus Aspergillus giganteus NTU96” written by Chen and co-workers describes on isolation of seven new highly oxygenated compounds along some known natural product including patulin and their biological activities. Since this paper seems important for drug discovery, the reviewer recommends this paper to be published in marine drugs. However, please mind the following points.
- All new compounds have small optical rotations. And the authors claim that compound 5 was deduced to be an racemate. How about others are? Natural patulin is apt to racemize from its hemiacetal structure. The authors should try to elucidate that by alternative method from optical rotation value. Chiral HPLC analyses or measurement of their CD spectra are recommended.
- Did the authors suspect compounds 3 and 4 might be artifact from patulin during isolation process. EtOAc was used in the first extraction and MeOH was used in re-solvation. So, it might be possible. The authors should confirm on that.
- Structures of compounds 3 and 4 are so close. How do the authors explain difference in bioactivities between them?
- Relation between compound 6 and 7 is interesting. Can they interconvert?
- Throughout the manuscript delta-s are not seen in the sent text.
Author Response
Responses to reviewer 1:
The manuscript entitled “Highly Oxygenated Constituents from A Marine Alga-derived Fungus Aspergillus giganteus NTU96” written by Chen and co-workers describes on isolation of seven new highly oxygenated compounds along some known natural product including patulin and their biological activities. Since this paper seems important for drug discovery, the reviewer recommends this paper to be published in marine drugs. However, please mind the following points.
- All new compounds have small optical rotations. And the authors claim that compound 5 was deduced to be an racemate. How about others are? Natural patulin is apt to racemize from its hemiacetal structure. The authors should try to elucidate that by alternative method from optical rotation value. Chiral HPLC analyses or measurement of their CD spectra are recommended.
Response: Patulin, like the glucopyranose, with an hemiacetate functionality will form a- and b-anomer simultaneously, which could not be separated by chiral HPLC to yield pure form. The CD spectra of compounds 1–7 have been acquired. As you can see below, no significant peak was observed in the CD spectra of all the tested compounds. Thus, it was speculated that all compounds 1–7 were racemates.
- Did the authors suspect compounds 3 and 4 might be artifact from patulin during isolation process. EtOAc was used in the first extraction and MeOH was used in re-solvation. So, it might be possible. The authors should confirm on that.
Response: In this study, the solid-sate fermented products underwent methanolic extraction initially, then open column separation by methanolic elution, and final HPLC purification with aqueous methanol as mobile phase. Considering all the C7 polyketides (1–7) and patulin isolated in this study, only patulin was found in the solid-state fermented products, while its methoxyl and ethoxyl analogues, aspergilsmins C–G (3–7), were isolated from the extract of submerged culture. Moreover, no aspergilsmins C–G (3–7) was found when pure patulin was dissolved either in methanol or ethyl acetate for several weeks. These data suggested all the methoxyl and ethoxyl derivatives of patulin would not be resulted from artifact.
- Structures of compounds 3 and 4 are so close. How do the authors explain difference in bioactivities between them?
Response: As described in lines 181–185, compound 4 bearing an ethoxyl group at its C-7 instead of a hydroxy and a methoxyl at C-7 of patulin and 3, respectively, reduced its bioactivity significantly. It was thus speculated that the size of the functional group attached at C-7 and an olefinic functionality at D4 of patulin could play crucial roles in the anti-cancer and anti-angiogenic activities.
- Relation between compound 6 and 7 is interesting. Can they interconvert?
Response: As shown in the NMR data of both 6 and 7 in the SI (Figure S27-S38), no mutual impurity was found, even after several weeks. Thus, no sign was observed for supporting that compounds 6 and 7 can be intercoverted.
- Throughout the manuscript delta-s are not seen in the sent text.
Response: The missing deltas could be resulted from font change, and have been revised.

Reviewer 2 Report
The paper “Highly Oxygenated Constituents from A Marine Alga-derived Fungus Aspergillus giganteus NTU967” describes the structure characterization seven oxygenated polyketides together with their bioactivities.
However, in my opinion, the manuscript is not enough well written to be accepted as well and many issues should be addressed.
First of all, during all the description of the purification author describe a fractionation guided by antimicrobial assay. However, in the end, the isolated compounds were tested (at least are this is what is reported) only for anticancer and anti-angiogenic functions. Did the authors evaluate antimicrobial activity? They say that they purify the active principle, which is?
The section “Extraction and Isolation of Secondary Metabolites” is confused. Authors should be much more clear in the description and could be useful report clearly the yield of each bioactive compounds in the two extracts, obtained from solid- and liquid-state culturing respectively.
The format is wrong, table and figure should be reported only after they citation in the text; some paragraphs in material and method should be rearranged. In all the text, when it is reported a chemical shift, the author should write δH and δC not only H or C.
Many other issues indicated directly on the text should be correct.

Author Response
The paper “Highly Oxygenated Constituents from A Marine Alga-derived Fungus Aspergillus giganteus NTU967” describes the structure characterization seven oxygenated polyketides together with their bioactivities.
However, in my opinion, the manuscript is not enough well written to be accepted as well and many issues should be addressed.
First of all, during all the description of the purification author describe a fractionation guided by antimicrobial assay. However, in the end, the isolated compounds were tested (at least are this is what is reported) only for anticancer and anti-angiogenic functions. Did the authors evaluate antimicrobial activity? They say that they purify the active principle, which is?
Response: In this study, the antimicrobial platform was just used as a preliminary screening and activity-directing tool for its efficiency and easy handling. All the separation process was guided by this assay, and the active principles, patulin and its analogues, were obtained finally. As described at results and discussion section at lines 73–76 in the MS: Patulin, a highly oxygenated C7 mycotoxin, with a hemiacetal functionality is apt to racemize naturally to form a pair of enantiomers, and was first characterized in 1943 under the name of tercinin as a potential antimicrobial agent. Because antimicrobial and anticancer assays were both cell-based platforms, they can be used to assess the similar cytotoxic characters of natural products to some extent. Thus, we evaluated the anticancer and antiangiogenic functions of the isolated compounds afterwards.
The section “Extraction and Isolation of Secondary Metabolites” is confused. Authors should be much more clear in the description and could be useful report clearly the yield of each bioactive compounds in the two extracts, obtained from solid- and liquid-state culturing respectively.
Response: All the yielded amounts of extracts and each compounds were described clearly at lines 216–246 in the MS.
The format is wrong, table and figure should be reported only after they citation in the text; some paragraphs in material and method should be rearranged. In all the text, when it is reported a chemical shift, the author should write δH and δC not only H or C.
Response: The missing deltas could be resulted from font change, and have been revised.
Many other issues indicated directly on the text should be correct.

Reviewer 3 Report
This manuscript describes the isolation of 7 polyketides, including several patulin-like compounds, as well as patulin itself and three other known metabolites. Structural elucidation of a 3-dihydrofuranone (1) and a γ-dihydropyranone (2) was made by a combination of NMR and MS analysis. The five other new compounds consisted of ring-open and acetalized variants of patulin and were determined by comparison with patulin. Biological data are presented for one of the new compounds plus patulin, although the text comments that all 11 isolates were tested. Patulin and its methyl acetal are shown to inhibit growth of prostate PC3 cells and hepatocellular carcinoma SK-Hep-1 cells (low micromolar activity), and these results were followed up by assays measuring the formation of capillary tubes as a surrogate for angiogenesis, and both were active. This is a nice study but some revision of the manuscript is necessary.
The NMR data should be carefully checked. The spectra appear sufficiently clean and demonstrate integrity with the proposed structures. For compound 1, the proton peaks at 2.60 and 2.62 ppm must be dd, as they are differentiated and therefore must couple with each other, as well as H-3. The description of these peaks in lines 89-90 should include the fact there are two, not one, methylene signals. Also in compound 1, C-3 is described as a carbinoyl peak; this is a poorly defined term, but typically it’s used for CHx-O-C(=O)-, which is not the situation here.
The COSY, HMBC and ROESY correlations in figure 2 should ideally be spread into a second figure, although use of colour would help clarify these correlations. A second figure for the ROESY data would be better, though, because it would allow the relative stereochemistry from the ROESY, such as in compounds 6 and 7, to be shown. Position numbering should be shown in Figure 2 to help clarify the text.
The concentration of sorafenib should be given.
Biological data for all the eleven isolates should be given. Cytotoxicity for all compounds should be presented in Table 3 at least (or, if preferred, in the Supporting Information); presumably only the active compounds were followed up in the capillary assay, so making the Figures more inclusive is not necessary if the data are not available.
There are some grammatic errors, such as plural/singular discrepancies, such as “advantage” in line 51 and “zones” in line 59, “substituted” in line 158; some sentences don’t scan and/or make sense, such as lines 55-60, line 74, lines 79-81.
Symbols are missing from the PDF generated version and should be checked in the proof, such as lines 89-94, 101-110, 116-135, 140, 142-166, 185, 272, 276.
References for the agar disc diffusion assays described in lines 55-60 should be included if this was previous work. Currently the meaning of the sentence is not apparent.
Author Response
This manuscript describes the isolation of 7 polyketides, including several patulin-like compounds, as well as patulin itself and three other known metabolites. Structural elucidation of a 3-dihydrofuranone (1) and a γ-dihydropyranone (2) was made by a combination of NMR and MS analysis. The five other new compounds consisted of ring-open and acetalized variants of patulin and were determined by comparison with patulin. Biological data are presented for one of the new compounds plus patulin, although the text comments that all 11 isolates were tested. Patulin and its methyl acetal are shown to inhibit growth of prostate PC3 cells and hepatocellular carcinoma SK-Hep-1 cells (low micromolar activity), and these results were followed up by assays measuring the formation of capillary tubes as a surrogate for angiogenesis, and both were active. This is a nice study but some revision of the manuscript is necessary.
The NMR data should be carefully checked. The spectra appear sufficiently clean and demonstrate integrity with the proposed structures. For compound 1, the proton peaks at 2.60 and 2.62 ppm must be dd, as they are differentiated and therefore must couple with each other, as well as H-3. The description of these peaks in lines 89-90 should include the fact there are two, not one, methylene signals. Also in compound 1, C-3 is described as a carbinoyl peak; this is a poorly defined term, but typically it’s used for CHx-O-C(=O)-, which is not the situation here.
Response: The methylene peaks at dH 2.60 and 2.62 were revised to be dH 2.61 (d, J = 6.2 Hz, H2-2) at line 89 as checked carefully again from HSQC experiment. The carbinoyl carbon was revised to be one oxygenated methine carbon at line 101.
The COSY, HMBC and ROESY correlations in figure 2 should ideally be spread into a second figure, although use of colour would help clarify these correlations. A second figure for the ROESY data would be better, though, because it would allow the relative stereochemistry from the ROESY, such as in compounds 6 and 7, to be shown. Position numbering should be shown in Figure 2 to help clarify the text.
Response: The COSY, HMBC, and ROESY of compound 5 were spread into 2 figure in Figure 2. COSY, HMBC, and ROESY were shown using different colors in Figure 2. Numbering of some critical positions was added.
The concentration of sorafenib should be given.
Response: The concentration of sorafenib (10 mM) was added in the legend of Figure 3.
Biological data for all the eleven isolates should be given. Cytotoxicity for all compounds should be presented in Table 3 at least (or, if preferred, in the Supporting Information); presumably only the active compounds were followed up in the capillary assay, so making the Figures more inclusive is not necessary if the data are not available.
Response: Detail cytotoxicity data of all the compounds were presented in the Table S1 of supporting information.
There are some grammatic errors, such as plural/singular discrepancies, such as “advantage” in line 51 and “zones” in line 59, “substituted” in line 158; some sentences don’t scan and/or make sense, such as lines 55-60, line 74, lines 79-81.
Response: The grammatic errors were revised according to suggestions.
Symbols are missing from the PDF generated version and should be checked in the proof, such as lines 89-94, 101-110, 116-135, 140, 142-166, 185, 272, 276.
Response: The missing deltas could be resulted from font change, and have been revised.
References for the agar disc diffusion assays described in lines 55-60 should be included if this was previous work. Currently the meaning of the sentence is not apparent.
Response: A reference was added at line 57.

Reviewer 4 Report
The manuscript "marinedrugs813805" deals with the isolation and identification of some new metabolites from liquid and solid fermented broth of Aspergillus giganteus NTU967. Furthermore, the antimicrobial and antitumor activity of the isolated compounds have been studied. In general, the manuscript reports interesting data concerning the isolation of new metabolites (whose structure determination is well discussed), one of these with promising antiproliferative activity, to be employed in drug development. But, in the present form this manuscript miss in some contents, that the authors describe in confusing way. According to what the authors wrote in the abstract and in the Introduction, the antimicrobial assay has been carried out on ethyl acetate extracts and it has showed promising results. These results are omitted in Result and Discussion section which starts with the spectroscopic characterization of isolated compounds, followed by anticancer activity studies. In Materials and Methods the authors state that ethyl acetate extracts were subjected to Sephadex LH-20 fractionation and the fractions (see lines 223 and 238) with the antimicrobial activity were further purified to give the pure compounds.
The authors have to complete this work discussing the omitted results in the proper section. In addition, the authors have to better define if extracts or fractions or both of them were subjected to the cited antimicrobial assay.
To the follow I report a list of other points to improve in the manuscript.
- I invite the authors to insert in Figure 1 the missing structures of known compounds identified as done for patulin.
- Please for deuterated solvent use the name methanol-d4 or CD3OD.
- Throughout the manuscript symbols (greek letters as δ) are missing.
- What about the anticancer and anti-angiogenic data of the other compounds, the authors should include also this values for completeness and to have further SAR information. In fact, at line 175 the authors wrote " As shown in Table 3, compound 3 and patulin exhibited most potent anti-angiogenic activities" the most potent respect to the other compounds probably, but how a reader can compare these values with those not reported? Again, at lines 182-187 the authors pointed out some structure-activity relationships, so it's important to give the data for all compounds examined.
- Please, improve the quality presentation, reorganise the figure position better, for instance, Figure 4 appears in Materials and methods, figure 3 is allocated before table 3, but it's commented later in the discussion.
- In General Experimental Procedures section the author should report the information of the 400 MHz NMR instrument too.
- Line 166 ROESY spectra suggested not confirmed the orientation of methoxy groups. Please use a different form than "phase" (for instance two groups can be oriented on opposite or on the same side).
Author Response
The manuscript "marinedrugs813805" deals with the isolation and identification of some new metabolites from liquid and solid fermented broth of Aspergillus giganteus NTU967. Furthermore, the antimicrobial and antitumor activity of the isolated compounds have been studied. In general, the manuscript reports interesting data concerning the isolation of new metabolites (whose structure determination is well discussed), one of these with promising antiproliferative activity, to be employed in drug development. But, in the present form this manuscript miss in some contents, that the authors describe in confusing way. According to what the authors wrote in the abstract and in the Introduction, the antimicrobial assay has been carried out on ethyl acetate extracts and it has showed promising results. These results are omitted in Result and Discussion section which starts with the spectroscopic characterization of isolated compounds, followed by anticancer activity studies. In Materials and Methods the authors state that ethyl acetate extracts were subjected to Sephadex LH-20 fractionation and the fractions (see lines 223 and 238) with the antimicrobial activity were further purified to give the pure compounds.
The authors have to complete this work discussing the omitted results in the proper section. In addition, the authors have to better define if extracts or fractions or both of them were subjected to the cited antimicrobial assay.
Response: In this study, the antimicrobial platform was just used as a preliminary screening and activity-directing tool for its efficiency and easy handling. All the separation process was guided by this assay, and the active principles, patulin and its analogues, were obtained finally. As described at results and discussion section at lines 77–70 in the MS: Patulin, a highly oxygenated C7 mycotoxin, with a hemiacetal functionality is apt to racemize naturally to form a pair of enantiomers, and was first characterized in 1943 under the name of tercinin as a potential antimicrobial agent. Because antimicrobial and anticancer assays were both cell-based platforms, they can be used to assess the similar cytotoxic characters of natural products to some extent. Thus, we evaluated the anticancer and antiangiogenic functions of the isolated compounds afterwards.
To the follow I report a list of other points to improve in the manuscript.
- I invite the authors to insert in Figure 1 the missing structures of known compounds identified as done for patulin.
Response: The structures of the isolated known compounds, deoxytryptoquivaline, tryptoquivaline, and quinadoline B were shown at Figure S39 of the SI.
- Please for deuterated solvent use the name methanol-d4 or CD3OD.
Response: The name of the deuterated solvent has been revised to be CD3OD.
- Throughout the manuscript symbols (greek letters as δ) are missing.
Response: The missing deltas could be resulted from font change, and have been revised.
- What about the anticancer and anti-angiogenic data of the other compounds, the authors should include also this values for completeness and to have further SAR information. In fact, at line 175 the authors wrote " As shown in Table 3, compound 3 and patulin exhibited most potent anti-angiogenic activities" the most potent respect to the other compounds probably, but how a reader can compare these values with those not reported? Again, at lines 182-187 the authors pointed out some structure-activity relationships, so it's important to give the data for all compounds examined.
Response: Detail cytotoxicity data of all the compounds were presented in the Table S1 of supporting information.
- Please, improve the quality presentation, reorganise the figure position better, for instance, Figure 4 appears in Materials and methods, figure 3 is allocated before table 3, but it's commented later in the discussion.
Response: The position of Figure 4 has been moved before “Materials and methods”. I tried to exchange Figure 3 and Table 3, but it seemed to be difficult. I think, if possible, they will be reorganized in the following process.
- In General Experimental Procedures section the author should report the information of the 400 MHz NMR instrument too.
Response: Bruker AVIII HD 400 was added at line 195.
- Line 166 ROESY spectra suggested not confirmed the orientation of methoxy groups. Please use a different form than "phase" (for instance two groups can be oriented on opposite or on the same side).
Response: The same phase was revised to be “the same side”.

Round 2
Reviewer 2 Report
The paper “Highly Oxygenated Constituents from A Marine Alga-derived Fungus Aspergillus giganteus NTU967” has been improved but, in my opinion, it still needs a strong reorganization to be acceptable for Marine Drugs.
In the abstract it is reported “Therefore, column chromatography of the active principles from liquid- and solid-state fermented products of the fungal strain was carried out, and which had led to isolation of eleven compounds.” However, in the section “Extraction and Isolation of Secondary Metabolites” (Material and Method), it seems that the new compounds were isolated only from liquid-state fermented products, whereas the known metabolites were purified from the solid-state fermentation. If this is true, the authors should explain clearly this concept in the manuscript.
Moreover, this sentence “Their structures were determined to be seven new highly oxygenated 29 polyketides, namely aspergilsmins A–G (1–7), along with previously reported patulin, 30 deoxytryptoquivaline, tryptoquivaline, and quinadoline B by spectral analysis” should be better expressed in English.
In the introduction, it is cited OSMAC approach but I didn’t understand the correlation with the work that has been done. Did the authors use it in this work? In case of a positive answer, it is not clear and should be reported; otherwise it is not functional to mention it.
Line 59: write were was collected the green alga Ulva lactuca
Line 65: move “2. Results and Discussion” after the figure
Line 67: delete “isolated in this study”
Line 72: specify known compounds isolated
Line 83: substitute “quasi-molecular ion” with pseudo-molecular ion”…”quasi” is usually used to indicate [M+H]+ not adduct with Na+
In section “2.2. Anti-cancer and Anti-angiogenic Assays of Secondary Metabolites” move the figure and the table below the text. Moreover, I suggest rearranging the graph with the biological data in only one figure.
Line 184: “ functionality at 4 of patulin” … at C4 of patulin
Section “Extraction and Isolation of Secondary Metabolites”: the terminology “portions” (e.g. line 222) is not appropriate…please change with samples.
Line 231: “For solid-state culture”… go to the head
Line 247, 252, 257, 261, 265, 269, 273: delete the number before the name of compounds. Check the format requested from the journal
Section from 3.4 to 3.8 (line 278-298): I suggest to write a section “biological assay” in which authors should also describe briefly the methods used (not only cited).
Author Response
The paper “Highly Oxygenated Constituents from A Marine Alga-derived Fungus Aspergillus giganteus NTU967” has been improved but, in my opinion, it still needs a strong reorganization to be acceptable for Marine Drugs.
In the abstract it is reported “Therefore, column chromatography of the active principles from liquid- and solid-state fermented products of the fungal strain was carried out, and which had led to isolation of eleven compounds.” However, in the section “Extraction and Isolation of Secondary Metabolites” (Material and Method), it seems that the new compounds were isolated only from liquid-state fermented products, whereas the known metabolites were purified from the solid-state fermentation. If this is true, the authors should explain clearly this concept in the manuscript.
Response: In this study, liquid-state and solid-state fermentation were both employed to enrich the chemical diversity of secondary metabolites of this fungal strain based on the concept of OSMAC, one strain and many compounds. In this study, all the new compounds were just purified from liquid-state culture. Actually, the results cannot be predicted. Sometimes, we also obtained new compounds only from solid-state culture.
Moreover, this sentence “Their structures were determined to be seven new highly oxygenated 29 polyketides, namely aspergilsmins A–G (1–7), along with previously reported patulin, 30 deoxytryptoquivaline, tryptoquivaline, and quinadoline B by spectral analysis” should be better expressed in English.
Response: This sentence has been revised to be “Their structures were determined by spectral analysis to be seven new highly oxygenated polyketides, namely aspergilsmins A–G (1–7), along with previously reported patulin, deoxytryptoquivaline, tryptoquivaline, and quinadoline B.”at lines 32-34.
In the introduction, it is cited OSMAC approach but I didn’t understand the correlation with the work that has been done. Did the authors use it in this work? In case of a positive answer, it is not clear and should be reported; otherwise it is not functional to mention it.
Response: OSMAC was a well-known strategy for microbial natural product research. In this study, we used both liquid- and solid-state culturing conditions with an attempt to enrich the chemical diversity of the fungal strain, and obtained highly oxygenated compounds from both samples finally. At lines 70-72, we have mentioned “ In this study, the green alga Ulva lactuca-derived fungal strain Aspergillus giganteus NTU967 was cultured in both solid- and liquid-state culturing conditions in order to enrich the diversity of the fungal secondary metabolites.”
Line 59: write were was collected the green alga Ulva lactuca
Response: the ethyl acetate extracts of fermented broths of Aspergillus giganteus NTU967 derived from the green alga Ulva lactuca were found to exhibit significant inhibition zone against S. aureus and C. neoformans. Because the “extracts” are plural, we used “were”.
Line 65: move “2. Results and Discussion” after the figure
Response: Revised.
Line 67: delete “isolated in this study”
Response: Revised.
Line 72: specify known compounds isolated
Response: At lines 73-74: “, patulin, deoxytryptoquivaline, tryptoquivaline, and quinadoline B,”was added.
Line 83: substitute “quasi-molecular ion” with pseudo-molecular ion”…”quasi” is usually used to indicate [M+H]+ not adduct with Na+
Response: Revised.
In section “2.2. Anti-cancer and Anti-angiogenic Assays of Secondary Metabolites” move the figure and the table below the text. Moreover, I suggest rearranging the graph with the biological data in only one figure.
Response: We integrated the original Fig.3 and Fig. 4 as the new_Fig.3 in this revised version for the demonstration of biological data. Furthermore, we have moved Table.3 and the new_Fig.3 below the text according your comment.
Line 184: “ functionality at 4 of patulin” … at C4 of patulin
Response: Revised.
Section “Extraction and Isolation of Secondary Metabolites”: the terminology “portions” (e.g. line 222) is not appropriate…please change with samples.
Response: Revised.
Line 231: “For solid-state culture”… go to the head
Response: Revised.
Line 247, 252, 257, 261, 265, 269, 273: delete the number before the name of compounds. Check the format requested from the journal
Response: Revised.
Section from 3.4 to 3.8 (line 278-298): I suggest to write a section “biological assay” in which authors should also describe briefly the methods used (not only cited).
Response: We have revised the description of biological assay in the Method section, and provided more detailed information for each assay used in anti-angiogenesis study according your comment.
